# Advancing Nanoscale Science: Synthesis and Bioprinting of Zeolitic Imidazole Framework-8 for Enhanced Anti-Infectious Therapeutic Efficacies

**DOI:** 10.3390/biomedicines11102832

**Published:** 2023-10-18

**Authors:** Muhammad Saqib Saif, Murtaza Hasan, Ayesha Zafar, Muhammad Mahmood Ahmed, Tuba Tariq, Muhammad Waqas, Riaz Hussain, Amna Zafar, Huang Xue, Xugang Shu

**Affiliations:** 1Faculty of Chemical and Biological Science, Department of Biochemistry, The Islamia University of Bahawalpur, Bahawalpur 63100, Pakistan; saqibsaif.qau@gmail.com (M.S.S.);; 2Faculty of Chemical and Biological Science, Department of Biotechnology, The Islamia University of Bahawalpur, Bahawalpur 63100, Pakistan; 3School of Chemistry and Chemical Engineering, Zhongkai University of Agriculture and Engineering, Guangzhou 510225, China; 4School of Engineering, Royal Melbourne Institute of Technology (RMIT) University, 24 La Trobe Street, Melbourne, VIC 3001, Australia; ayeshazafar510@gmail.com; 5Faculty of Chemical and Biological Science, Department of Bioinformatics, The Islamia University of Bahawalpur, Bahawalpur 63100, Pakistan; mahmoodahmad@iub.edu.pk (M.M.A.);; 6Faculty of Chemical and Biological Science, Department of Veterinary Sciences, The Islamia University of Bahawalpur, Bahawalpur 63100, Pakistan

**Keywords:** metal organic frameworks MOFs, nanocomposite, CME@ZIF-8, antibacterial

## Abstract

Bacterial infectious disorders are becoming a major health problem for public health. The zeolitic imidazole framework-8 with a novel *Cordia myxa* extract-based (CME@ZIF-8) nanocomposite showed variable functionality, high porosity, and bacteria-killing activity against *Staphylococcus aureus,* and *Escherichia coli* strains have been created by using a straightforward approach. The sizes of synthesized zeolitic imidazole framework-8 (ZIF-8) and CME@ZIF-8 were 11.38 nm and 12.44 nm, respectively. Prepared metal organic frameworks have been characterized by gas chromatography–mass spectroscopy, Fourier transform spectroscopy, UV–visible spectroscopy, X-ray diffraction, scanning electron microscopy, and energy-dispersive X-ray spectroscopy. An antibacterial potential comparison between CME@ZIF-8 and zeolitic imidazole framework-8 has shown that CME@ZIF-8 was 31.3%, 28.57%, 46%, and 47% more efficient than ZIF-8 against *Staphylococcus aureus* and 43.7%, 42.8%, 35.7%, and 70% more efficient against *Escherichia coli,* while it was 31.25%, 33.3%, 46%, and 46% more efficient than the commercially available ciprofloxacin drug against *Staphylococcus aureus* and 43.7%, 42.8%, 35.7%, and 70% more efficient against *Escherichia coli,* respectively, for 750, 500, 250, and 125 μg mL^−1^. Minimum inhibitory concentration values of CME@ZIF-8 for *Escherichia coli* and *Staphylococcus aureus* were 15.6 and 31.25 μg/mL respectively, while the value of zeolitic imidazole framework-8 alone was 62.5 μg/mL for both *Escherichia coli* and *Staphylococcus aureus*. The reactive oxygen species generated by CME@ZIF-8 destroys the bacterial cell and its organelles. Consequently, the CME@ZIF-8 nanocomposites have endless potential applications for treating infectious diseases.

## 1. Introduction

Since the development of antibiotic resistance, pathogenic diseases caused by bacteria have gained significant public health interest [1]. Antimicrobial resistance is increasing every day, and microbes’ ability to defeat the drugs has become an alarming situation [2]. A report from the World Health Organization (WHO) urges swift, concerted, and aspiring action to prevent a horrific antimicrobial-resistance tragedy. If no action is taken until 2050, drug-resistant illnesses might cause the death of 10 million people annually [3]. At least 700,000 people in a year die from drug-resistant infirmity, and nations invest considerably in cutting-edge science and technology to contend with resistance against antibiotics [4]. A Gram-positive bacterium *Staphylococcus aureus* (MRSA) is resistant to the drug methicillin, and it infects individuals globally and has become the cause of many diseases, i.e., pneumonia, renal failure, blood poisoning, and toxic shock syndrome [5]. In accordance with the Center for Disease Control and Prevention, Gram-negative bacteria are resistant to third-generation cephalosporin [6]. Malathy Iyer discovered that *Escherichia coli* (mutant strain) became resistant to carbapenem (imipenem/cilastatin) [7]. Extended-spectrum β-lactamase (ESBLs) *E. coli* pathogenicity genes contribute to the development of UTIs, kidney disorders, and neonatal encephalitis [8,9]. Antibacterial agents can efficiently pass through the thick but porous cell walls of Gram-positive bacteria, which are composed of peptidoglycan (20–80 nm) formed of teichoic acid, glucosamine, and N-acetylated muramic acid and have a high negative charge, while Gram-negative strains are composed of two layers of membranes with negatively charged oligosaccharides and lipoproteins at 5–10 nm; therefore, a substantial number of new antibacterial drugs are thus being produced to decrease the overuse of antibiotics [10]. Recently, nanomaterials (1–100 nm) have become an effective substitute tool for tackling multidrug-resistant microorganisms [11].

The physical and chemical properties of nanomaterials offer a flexible base for developing new therapeutic approaches for microorganisms with antibiotic resistance [12]. According to recent works, it has been found that oxidative stress in cells produced by ROS harms the organelles of bacterial cells. Different groups of nucleic acids and proteins like [(–SH), (–NH), (–COOH)] interact with nanoparticles, disrupting the enzymes, altering cell structure, and inhibiting the microbe [13]. Many different therapeutic compounds can be produced by plants and, additionally, can be employed as biomaterials. *Cordia myxa* (sapistan) is a profoundly known herbal medicine that is anti-inflammatory and analgesic, antioxidant, immunomodulatory, anti-stomach ulcer, potential-reducing, blood pressure-controlling, and antiparasitic [14,15]. Numerous plant metabolites, including alkaloids, terpenoids, coumarins, phenolic acids, tannins, flavonoids, sterols, and saponins are abundant in *C. myxa*, making them suitable for human consumption [16].

A class of metal–organic framework, ZIF-8 is formed of 2-methylimidazole and zinc ions, with a sodalite topological crystal in a cubic form and with a lattice constant of 1.61 nm. Large molecules cannot fit through the pores because ZIF-8’s pore-opening diameter is only 3.4 Å (0.34 nm), while 11.6 Å (1.16 nm) is the diameter of its pore-cavity. When exposed to water and organic solvents for seven days at 50 degrees, it shows a high temperature permanence of up to 550 °C in N2 conditions and no structural deterioration [17]. ZIF-8 is becoming more and more important for use in thin-film devices [18], electrochemistry [19], bioimaging [20], and drug delivery, and it can also be used as a storage medium like methane and hydrogen gasses [21] and as a highly efficient adsorbent [22].

Different researchers worked on different nanomaterials to check their potential as antibacterial agents. Chitosan-coated NMOFS were synthesized with the loading of vancomycin to check its efficiency against the *S. aureus* bacterial strain. The inhibition index values against the *S. aureus* demonstrates that it is more bioactive, and it has a higher catalytic property [23]. Another research based on the synthesis of the three-type Zn metal incorporated nMOFs like (IRMOF-3), (MOF-5), and (Zn-BTC) and checked against four different bacterial strains. Different concentrations of MOF and a conjugated drug-based MOF showed different antibacterial activities. It showed that the ZN-based MOF alone cannot work efficiently while its catalytic index increases with drug combination therapy [24]. Co-MOF (UoB-3) was synthesized and checked against *E. coli* and *Bacillus cereus* and had maximum potential towards bacterial killing [25]. The Ag-NPs@Ni-MOF organic nanostructure was developed, and its antibacterial efficacy was calculated. The calculated inhibitions showed that upon incorporation of Ag, the potency of antibacterial activity was raised [26].

In this work, the ZIF-8 nanostructure was fabricated, along with the addition of stabilizing and capping the phytochemical metabolites from the *Cordia myxa* plant to increase its bioactivity (dubbed CME@ZIF-8). We aimed to synthesize the cost-effective nanomaterial from medicinal plants with bioactive elements and to check its efficacy as an antimicrobial agent by designing a novel MOF (ZIF-8). Multiple characterizations were performed, such as UV–visible spectrophotometry, XRD crystallography, energy-dispersive X-ray (EDX) spectroscopy, scanning electron microscopy (SEM), and Fourier-transform infrared (FTIR) spectroscopy, to confirm the formation of the ZIF-8 nanocomposite. Antimicrobial activity was performed against *S. aureus* and *E. coli*. By comparing their results, it was analyzed that the ZIF-8 nanocomposite was more effective against the bacterium, as compared to ciprofloxacin, CME, and ZIF-8. It is also possible that combination therapy will be the key element of upcoming studies.

## 2. Materials and Methods

### 2.1. Materials

Zinc acetate hexahydrate (Sigma-Aldrich, Merck KgaA, Darmstadt, Germany), 2-methyl imidazole (Sigma-Aldrich), nutrient broth (Sigma-Aldrich), Luria-Bertani agar (Sigma-Aldrich), iodonitrotetrazolium dye ciprofloxacin, ethanol, methanol, and distilled water, were commercially available and used without any further purification.

### 2.2. Plant Extract Preparation

To make the plant extract, we used 25 g of powder (*C. myxa* whole plant, collected from native areas of Bahawalpur) in a beaker with 500 mL of distilled water. For 2 h, the mixture was heated at 80 °C. The *C. myxa* extract (CME) was filtered with Whatman filter paper and was left at 80 °C in an oven to completely remove the water, and the dry powder form extract was stored for further use.

### 2.3. Synthesis of ZIF-8

Following the method created by Harpreet Kaur and colleagues, ZIF-8 was created in a completely aqueous solution at room temperature [27]. The reagents were zinc acetate and 2-methylimidazole (2 MeIM, with a metal-to-ligand ratio of 1:4). An amount of 0.219 g of metal acetate and 0.328 g of 2-MeIM were blended in 10 mL of distilled water in a separate flask. The solution of zinc salt was combined with the 2 MeIM solution dropwise under vigorous stirring for 1 h. White precipitates were formed as the product, as shown in Figure 1.

### 2.4. CME@ZIF-8 Synthesis

CME@ZIF-8 was synthesized using a method consistent with prior research [4]. Specifically, 0.219 g of zinc acetate hexahydrate was mixed in 10 mL of distilled water, while 0.328 g of 2-methylimidazole was blended in 20 mL of distilled water, along with 0.5 g of CME. The mixture of CME and 2-methylimidazole was stirred for 15 min before the gradual addition of the zinc acetate solution. Within 3 to 4 min, the brown reaction solution underwent a color change, turning creamy in appearance, representing the makeup of the CME@ZIF-8 nanocomposite. Following this, the CME@ZIF-8 nanocomposite solution was left at room temperature for 24 h. To obtain the precipitate of the CME@ZIF-8 nanocomposite, the solution was centrifuged at 10,000× *g* rpm for a spell of 10 min. Successively, it experienced three rounds of washing with 10 mL of distilled water each time to remove any residual unreacted chemicals. The resulting material was then dried at 80 °C. Following this, both the CME@ZIF-8 nanocomposite and ZIF-8 were crushed into a powder using a mortar and pestle, as depicted in Figure 1.

### 2.5. Characterization

The absorption spectra of both the synthesized CME@ZIF-8 nanocomposite and ZIF-8 were measured between a 200–800 nm range by using a spectrophotometer (Epoch-BioTek Instruments, USA) with a resolution set at 1 nm. Gas chromatography–mass spectrometry (GC-MS) analysis was employed to detect the chemical compounds (GCMS-QP2010 Plus). The FTIR absorption spectra were obtained by using KBr pellets with an Agilent Technologies FT-IR spectrometer. The spectra were scanned over a 4000 to 650 cm^−1^ range at a scanning speed of 2 cm^−1^. This technique was accomplished to identify and characterize the different functional groups present in the samples. The XRD patterns of the samples were noted using a Bruker AXS diffractometer (D8 Advance). A copper Kα source (wavelength = 1.542 Å) was utilized, with X-ray generation set at 40 kV and 35 mA. The scan range for the XRD analysis was from 5° to 50°. Scanning electron micrographs (SEMs) were taken on an A JEOL JSM 6610 scanning electron microscope. Energy-dispersive spectrophotometer (EDS) was employed in combination with scanning electron microscopy (SEM) to investigate the elemental compositions of the samples.

### 2.6. Antimicrobial Tests

#### 2.6.1. Disc Diffusion Method

We used the Gram-positive strain *S. aureus* and the Gram-negative strain *E. coli* as model organisms (obtained from the Pathology Department, BVH Hospital). Bacteria were cultured in nutrient broth media in a shaking incubator overnight at 37 °C and 140 rpm. Bacterial cultures were diluted to reach the OD between 0.4 to 0.6 at 600 nm. To check the antibacterial activity of the nanocomposites, the bacterial culture density was further decreased to 10^5^ CFU mL^−1^. A total of 100 μL of culture was spread on the plates on Luria-Bertani agar, and various doses of ZIF-8, CME@ZIF-8, CME, and ciprofloxacin (750, 500, 250, and 125 μg mL^−1^) were applied to 6 mm diameter disks. The disks were applied on agar plates and placed in the oven for overnight incubation at 37 °C. The next day, zones of inhibition were measured to determine the antibacterial potentials of the samples.

#### 2.6.2. Minimum Inhibitory Concentration

The minimum inhibitory concentration (MIC) test was examined as per the method used previously by [28]. In this study, CME@ZIF-8, ZIF-8, CME, and ciprofloxacin were used against *S. aureus* and *E. coli.* The experiments were conducted using 96-well test plates. Each well was initially loaded with 100 mL of nutrient broth, followed by the addition of the samples via serial dilution (250 μg to 0.244 μg). After that, an additional 30 mL of fresh bacterial culture was introduced to the wells as part of the experimental procedure. Plates were placed in the incubator at 37 °C for 24 h. After incubation, 50 mL of INT dye solution (1 mg/mL) prepared in methanol was added into each well, and then plates were incubated again at 37 °C for 30 min. Colorimetric visualization was used to assess the growth and inhibition. The MIC of each sample was represented by uncolored wells.

## 3. Results and Discussion

In accordance with the referenced literature [1], we successfully encapsulated CME within the ZIF-8 framework, resulting in the formation of CME@ZIF-8. This encapsulation process involves the addition of an organic linker, which leads to the disassembly of metal ions within the coordination polymer. Subsequently, when these metal ions and linkers reassemble, ZIF-8 crystals are formed. In our study, we achieved the creation of hierarchical CME@ZIF-8 by incorporating the target molecules within the ZIF-8 synthesis process. Figure 1 illustrates that various biomolecules with diverse functional groups were effectively encapsulated into the ZIF-8 crystals.

### 3.1. Physico-Chemical Characterizations of Prepared MOFs

To identify the bio-reducing chemical compounds responsible for the antibacterial action and to confirm the encapsulation of CME into ZIF-8 during water-based extraction, a comprehensive analysis was conducted. This analysis involved comparing the retention times and mass/weights of the compounds with genuine standard samples using gas chromatography (GC) and mass spectra from reputable databases, including the Wiley Libraries, ChemSpider, Royal Society of Chemistry, and PubChem (NIH). These methods were employed to ensure the accurate identification and characterization of the chemical constituents and to validate the encapsulation of CME within the ZIF-8 framework.

The identified compounds within CME and CME@ZIF-8 with their molecular formulas and molecular weights are listed in Table 1 and Table 2, respectively. The chromatogram of CME is shown in Figure 2a, while CME@ZIF-8 is shown in Figure 2b. It was observed that hexadecenoic acid and methyl ester, methyl (9E,12E)-9,12-octadecadienoate, oleic acid and methyl ester, and stearic acid and methyl ester were the compounds that were also found in the GC-MS analysis of the CME@ZIF-8, which affirmed the incorporation of the CME into the CME@ZIF-8. These chemicals had antibacterial potential, as mentioned in Table 1 and Table 2, so they increased the antimicrobial potential of the CME@ZIF-8.

We verified coordinated CME@ZIF-8 with Zn^2+^ ions using the UV−vis spectrum and compared the CME@ZIF-8 with the ZIF-8 and CME spectra, as shown in Figure 3a. No absorption peak was seen for ZIF-8. CME exhibited peaks at 210, 225, and 275 nm, which corresponded to saponins, phenols, and flavonoids. When CME was encapsulated within ZIF-8, CME@ZIF-8 displayed two absorption peaks. The peak at 216 nm corresponded to ZIF-8 [27], while the peak at 230 nm indicated the presence of phenolic compounds from CME [47]. This observation confirms the successful encapsulation of CME into ZIF-8. The encapsulation process likely involves the formation of intermolecular H-bonds linking the phenolic OH-group of CME and the “N” atoms in 2-MeIM, which facilitates the encapsulation process [4].

Moreover, the existence of different functional groups within the CME@ZIF-8 nano-bio-composite, CME, and ZIF-8 were confirmed by FTIR, as shown in Figure 3b. For the ZIF-8, four absorption peaks at 995, 1145, 1577, and 2926 cm^−1^ showed C–N, C=N, and C–H groups in the imidazole ring, respectively [48]. In the free *C. myxa* extract, absorption peaks at 1035, 2028, 2904, and 3263 showed C–O, X=C=Y, C–H, and O–H functional groups, and for CME@ZIF-8, peaks at 1035, 2028, 2904, and 3390 were assigned to vibrations for C–O, X=C=Y, C–H, and O–H functional groups, which were the same as CME and revealed the encapsulation of the CME into the ZIF-8, confirming the formation of the nano-bio-composite. The peak at 1583 cm^−1^ in CME@ZIF-8 was attributed to the C=C–C stretch of the aromatic group ring. Additionally, CME@ZIF-8 exhibited a planar bent vinyl C-H absorption peak at 1420 cm^−1^. In CME@ZIF-8, there were also characteristic peaks at 760 cm^−1^, associated with the aromatic C-H stretch, and at 919 cm^−1^, corresponding to the vinyl terminal absorption. These spectroscopic findings align with previous research that has been published and have been shown to yield more robust and reliable results [48].

The XRD analysis indicated that CME@ZIF-8 particles exhibited a high degree of crystallinity. The presence of CME within the pores of ZIF-8 crystals resulted in broader peaks for CME@ZIF-8. However, the peaks observed for both ZIF-8 and CME@ZIF-8 closely matched the simulated XRD pattern for ZIF-8 and its cubic unit cell. This alignment was confirmed by the crystallographic database (JCPDS 00-062-1030).

The XRD patterns for ZIF-8 and APE@ZIF-8 were observed at 2θ angles of 7.34, 10.37°, 12.26°, 13.96°, 14.91°, 17.24°, 18.52°, 23.17°, 27.32°, and 29.76° and at 10.44°, 12.48°, 14.24°, 15.43°, 18.05°, 22.76°, 26.96°, and 29.58°. These diffraction peaks are consistent with previous studies [49,50] and correspond to the crystallographic planes of (011), (002), (112), (022), (013), (222), (233), (134), (044), and (244), respectively, as shown in Figure 4.

The crystallite sizes of the grown MOFs samples were calculated by using the Scherrer equation, given as [28]:$$D=\frac{K\lambda }{\beta \cos\theta }$$

The average crystallite size of MOFs is the *D*, *K* is the Scherrer constant, *λ* is the wavelength of the source of the *X*-ray, Bragg’s angle is *θ*, and *β* is the line broadening at FWHM [51]. The average crystallite size of ZIF-8 MOFs was 11.38 nm, as shown in Appendix A, and the average size of *C. myxa*-based CME@ZIF-8 MOFs was 12.44 nm, as shown in Appendix A. The current work is supported by [52], which synthesized ZIF-8 between 8 and 33 nm. The CME@ZIF-8 size was higher than ZIF-8 because of the encapsulation of CME in ZIF-8 crystals. According to SEM, the morphology of pure ZIF-8 NPs was like platelets, as shown in Figure 5a, and according to SEM, images of CME@ZIF-8 revealed a layered surface structure, which is different from ZIF-8 crystals, which have a smooth surface, as shown in Figure 5b. A comparison between these images also approved the surface and porous attachments of biomolecules [53].

Based on the histogram analysis, the average particle sizes of CME@ZIF-8 and ZIF-8 were determined using SEM. In Figure 5c, the particle size of ZIF-8 was measured to be 409 nm (average), while in Figure 5d, the size of CME@ZIF-8 was found to be 732 nm (average). These results matched with previously synthesized ZIF-8 MOFs [54]. The measured sizes from the SEM analysis were greater than the crystallite sizes calculated from the XRD. This was because the grain or particle was formed by the aggregation of a number of nanocrystals [55]. Appendix A displays the EDXs of the CME@ZIF-8 nanocomposite and pure ZIF-8. For CME@ZIF-8, a 41.4% weight for carbon, 29.5% weight for nitrogen, 8.2% weight for oxygen, and 20.9% weight for zinc were measured. For pure ZIF-8, a 41.4% weight for carbon, 28.3% weight for nitrogen, 7.7% weight for oxygen, and 22.5% weight for zinc were measured. The presence of C, N, O, and Zn upon the surface of the prepared organic framework and the lack of any impurities were verified by EDX analysis [56]. Moreover, there were comparative relationships among the values of the components already present in the sample and the peak intensities of each element. These findings highlight the high purity of the synthesized nanomaterials and the close agreement between their compositions and the suggested mass percentages [57].

### 3.2. Antibacterial Activity of MOFs

#### 3.2.1. Disc Diffusion Method

Various concentrations of pure ZIF-8, CME, and the standard drug ciprofloxacin were employed to compare the antibacterial activities with the CME@ZIF-8 nanocomposite. These concentrations were set at 750, 500, 250, and 125 μg/mL in distilled water for both *S. aureus* and *E. coli* bacteria. Across all concentrations, the CME@ZIF-8 NPs exhibited the highest zones of inhibition (ZOI) against both bacterial species, followed by ciprofloxacin, ZIF-8, and CME.

Figure 6a,b displays the values of ZOI for CME@ZIF-8, ZIF-8, CME, and ciprofloxacin, revealing varying levels of bacterial killing potential against the two bacterial strains. Following the incubation of bacteria with CME@ZIF-8 NPs, it was evident that the ZOIs were exceptionally large. On the other hand, ZIF-8 and CME exhibited relatively limited antibacterial activity, although this activity improved when they were combined or conjugated with CME@ZIF-8 NPs [4]. The different plant extract concentrations showed bactericidal behavior against *S. aureus* (13 mm, 11 mm, 0 mm, and 0 mm) and against *E. coli* (14 mm, 11 mm, 0 mm, and 9 mm); in both cases, 750 μg/mL showed the maximum result. The antibacterial activities of CME@ZIF-8 MOFs with different concentrations (750, 500, 250, and 125 μg mL^−1^) offered higher antibacterial activities against *S. aureus* (22 mm, 20 mm, 20 mm, and 18 mm) and antibacterial activities against *E. coli* (30 mm, 26 mm, 24 mm, and 24 mm). The antibacterial activities of ZIF-8 NPs with different concentrations (750, 500, 250, and125 μg mL^−1^) offered higher antibacterial activities against *S. aureus* (16 mm, 14 mm, 12 mm, and 12 mm) and antibacterial activities against *E. coli* (16 mm, 14 mm, 14 mm and 10 mm), respectively; in these cases, 750 ug/mL also showed the maximum result. Additionally, the ciprofloxacin control showed antibacterial activities against *S. aureus* (16 mm, 15 mm, 12 mm, and 10 mm) and against *E. coli* (20 mm, 18 mm, 15 mm, and 13 mm).

A comparison between CME@ZIF-8 and ZIF-8 showed that CME@ZIF-8 was 31.3%, 28.57%, 46%, and 47% more efficient than ZIF-8 against *S. aureus,* while it was 43.7%, 42.8%, 35.7%, and 70% more efficient against *E. coli* for 750, 500, 250, and 125 μg mL^−1^, as shown in Figure 6b. CME@ZIF-8 was 46.75%, 40%, 20%, and 18% more efficient than CME (*C. myxa* extract) against *S. aureus,* while it was 57.14%, 68.18%, 24%, and 83.3% more efficient against *E. coli* for 750, 500, 250, and 125 μg mL^−^1. CME@ZIF-8 was 31.25%, 33.3%, 46%, and 46% more efficient than the commercially available ciprofloxacin drug against *S. aureus* and *E. coli,* respectively, for 750, 500, 250, and 125 μg mL^−1^.

As the concentration of CME@ZIF-8 increased, there was a noticeable increase in bactericidal potential. Notably, CME@ZIF-8 exhibited significantly varying bactericidal effects against *S. aureus* and *E. coli*, likely due to alterations in the cell wall structure, as depicted in Figure 7a,b. It is anticipated that CME@ZIF-8 NPs adhere to and penetrate the bacterial cell membrane to disrupt it. This is achieved through electrostatic attraction between the positively charged surfaces of the nanoparticles and the negatively charged surfaces of the bacterial cells.

Bacterial cells are believed to have pores with sizes ranging from 5 to 50 nm [58]. In contrast, the synthesized MOFs typically have sizes between 10 and 15 nm. Consequently, CME@ZIF-8 NPs can readily penetrate the cell walls of *S. aureus* and *E. coli*, leading to the release of intracellular components and probably causing cell lysis. This interference with the translocation process involved in tRNA production inhibits protein formation. Additionally, cell wall disruption occurs, due to cation displacement, facilitating the binding of lipopolysaccharides to other molecules. Oxidative stress induced by the generation of ROS leads to the destruction of the cell membrane, DNA, and proteins. This process results in the early-stage killing of bacteria and ultimately leads to bacterial cell death [59]. The findings from this study highlight that CME@ZIF-8 has the potential to serve as a potent nano-bacterial agent, due to its ability to induce such oxidative stress and effectively combat bacterial infections.

#### 3.2.2. Minimum Inhibitory Concentration (MIC)

When no observable growth was evident in the 96-well microtiter plates post-treatment, the MIC was reported. In the case of CME@ZIF-8 NPs, the MIC values for *S. aureus* and *E. coli* were determined to be 31.25 g/mL and 15.62 g/mL, respectively. CME@ZIF-8 NPs demonstrated the lowest concentration required for inhibition when compared to ciprofloxacin, ZIF-8, and CME against both bacterial strains.

The MIC values of CME@ZIF-8, ZIF-8, CME, and the positive control standard medicine ciprofloxacin were significantly lower for *E. coli* than that for *S. aureus,* as show in Figure 8a,b. MIC values of ciprofloxacin for *E. coli* and *S. aureus* were 31.5 and 62.5 μg/mL, respectively, and the values of *C. myxa* extract (CME) were 62.5 and 125 μg/mL, respectively. MIC values of CME-based synthesized CME@ZIF-8 for *E. coli* and *S. aureus* were 15.6 and 31.25 μg/mL, respectively, while values of ZIF-8 MOFs alone were the same for *E. coli* and *S. aureus* at 62.5 μg/mL. Figure 8c indicates that biologically synthesized CME@ZIF-8 MOFs showed 75% and 50% higher inhibitions when compared to simple ZIF-8 for *E. coli* and *S. aureus,* respectively. The results of our study reveal that CME@ZIF-8 MOFs produced through biological synthesis exhibit greater efficacy against bacterial strains, compared to standard ZIF-8 MOFs [4]. These findings suggest that using CME@ZIF-8 on bacteria strains such as *E. coli and S. aureus* could reduce the need for high antibiotic doses and, consequently, lessen the bacterial resistance resulting from excessive and non-standard antibiotic usage.

## 4. Conclusions

In conclusion, we successfully developed the nano-antibacterial agent CME@ZIF-8 by encapsulating bioactive compounds from the *Cordia myxa* plant extract in a single step into ZIF-8. CME@ZIF-8 can be a strong antibacterial agent, and it has exceptional biocompatibility, due to the Cordia myxa plant metabolites, as confirmed by our results. Experimental work revealed that CME@ZIF-8 can stop *S. aureus* and *E. coli* bacteria from growing. CME@ZIF-8 and ZIF-8 showed promising antibacterial activity against bacterial species. The Cordia myxa-encapsulated nanomaterials showed the highest inhibition index. This study will pave a pathway for developing a novel MOF structure (ZIF-8 and CME@ZIF-8) for biological application.

## Figures and Tables

**Figure 1 biomedicines-11-02832-f001:**
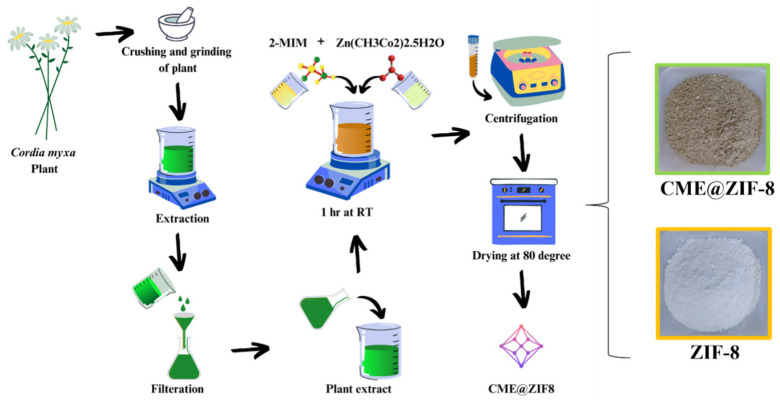
Schematic illustration of the CME@ZIF-8 nano-bio-composite synthesis for the resistant infectious agent.

**Figure 2 biomedicines-11-02832-f002:**
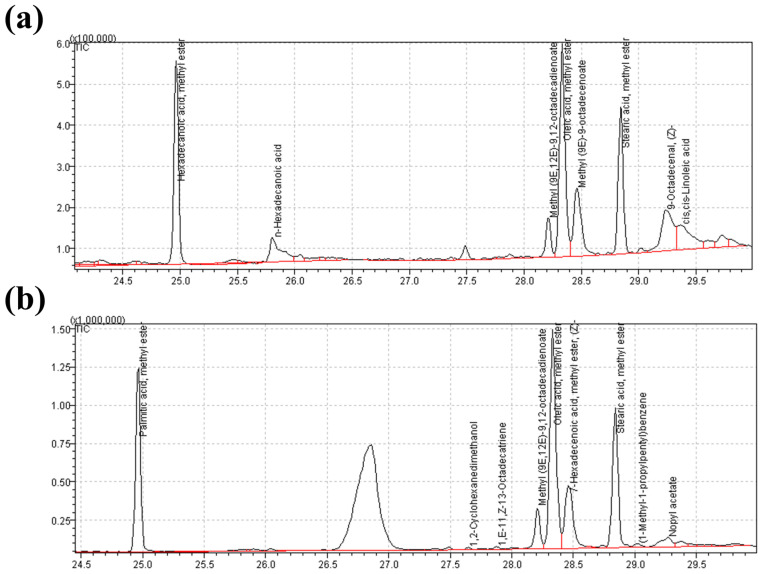
GC-MS chromatogram of bioactive molecules; (**a**) *C. myxa* crude extract; (**b**) CME@ZIF-8.

**Figure 3 biomedicines-11-02832-f003:**
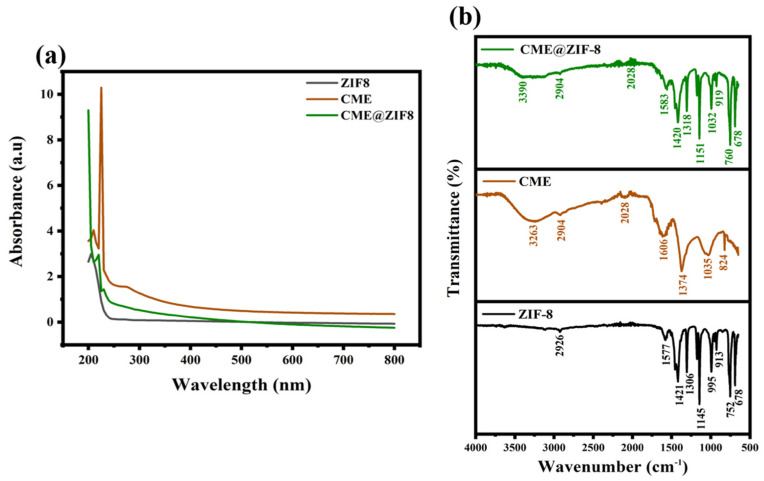
(**a**) UV–vis plot; (**b**) FTIR spectra of ZIF-8, CME, and CME@ZIF-8.

**Figure 4 biomedicines-11-02832-f004:**
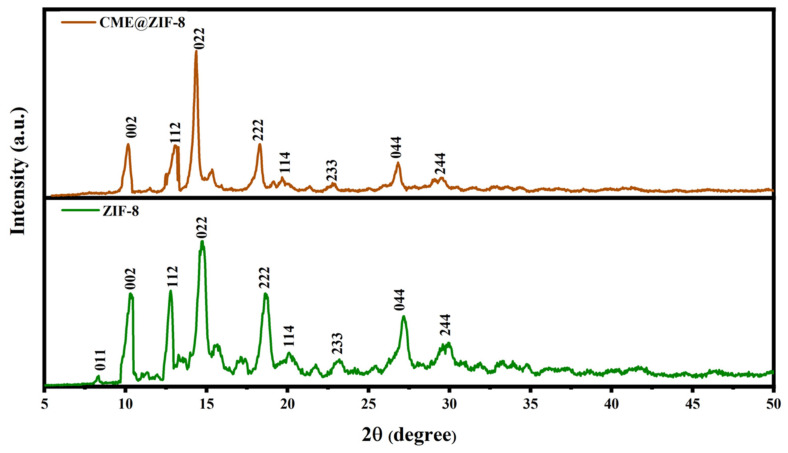
XRDs of ZIF-8 and CME@ZIF-8 for confirmation of loading.

**Figure 5 biomedicines-11-02832-f005:**
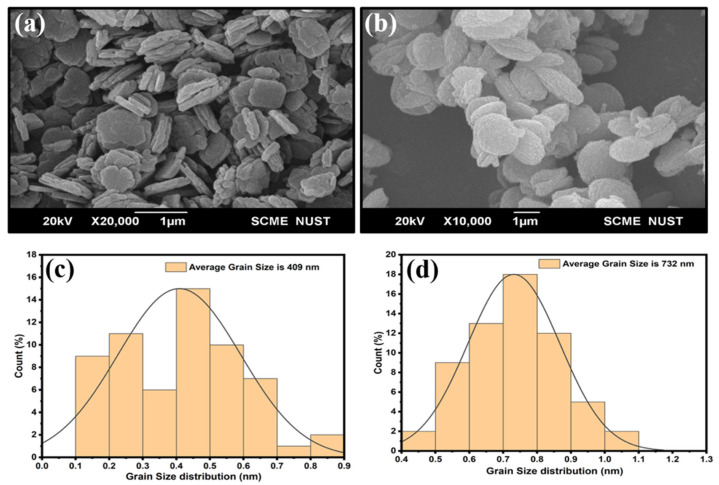
SEMs of (**a**) ZIF-8 and (**b**) CME@ZIF-8. Grain size distribution of (**c**) ZIF-8 and (**d**) CME@ZIF-8.

**Figure 6 biomedicines-11-02832-f006:**
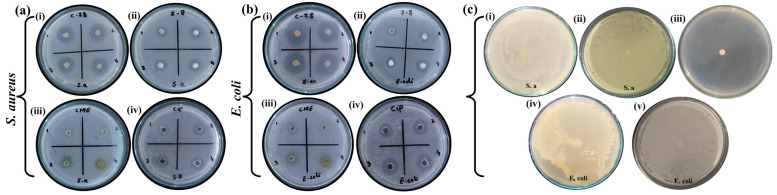
Typical disc diffusion pictures of ZOI in (**a**) *S. aureus* and (**b**) *E. coli* after treatment. (i) CME@ZIF-8, (ii) ZIF-8, (iii) CME, and (iv) ciprofloxacin; (**c**) (i) *S. aureus* without disc, (ii) untreated disc with *S. aureus,* (iii) blank, (iv) *E. coli* without disc, (v) untreated disc with *E. coli*.

**Figure 7 biomedicines-11-02832-f007:**
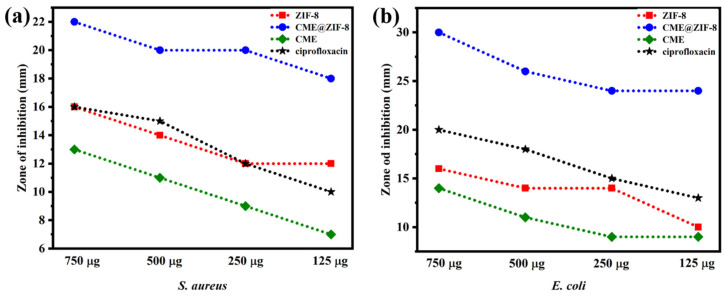
Zones of inhibition of ZIF-8, CME@ZIF-8, CME, and ciprofloxacin in (**a**) *S. aureus* and (**b**) *E. coli* shown on the histogram.

**Figure 8 biomedicines-11-02832-f008:**
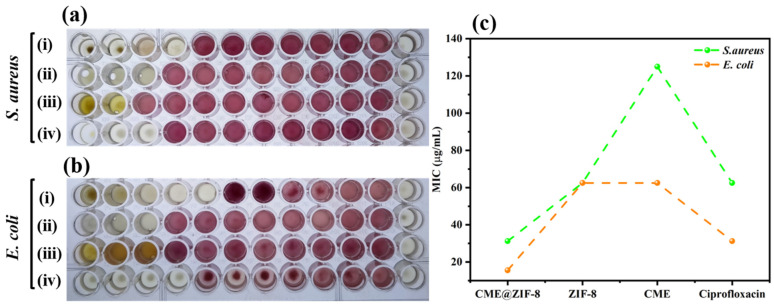
MICs of ciprofloxacin against (**a**) *S. aureus* and (**b**) *E. coli*. (**c**) MIC values against *S. aureus* and *E. coli.* (i) CME@ZIF-8, (ii) ZIF-8, (iii) CME, and (iv) ciprofloxacin.

**Table 1 biomedicines-11-02832-t001:** Summary of the properties, formulas, molecular weights, and retention times for significant compounds identified in the *C. myxa* crude extract.

No	RT (m)	Area	Compound Name	MF	Mol. Wt. g/mol	Properties
1	15.115	4651	3,8-dimethylundecane	C_12_H_26_	170.33	Antibacterial [29]
2	23.817	9778	Undecanoic acid,10-methyl-, methyl ester	C_13_H_26_O_2_	214.34	Anticancer [30]
3	24.965	279,571	Hexadecanoic acid, methyl ester	C_17_H_34_O_2_	270.5	Antibacterial [31]
4	25.805	16,632	n-Hexadecanoic acid	C_16_H_32_O_2_	256.42	Antimicrobial [32], antioxidant, and anti-inflammatory [33]
5	28.213	24,517	Methyl (9E,12E)-9,12-octadecadienoate	C_19_H_34_O_2_	294.5	Antibacterial [34], Antioxidant [35]
6	28.334	127,970	Oleic acid, methyl ester	C_19_H_36_O_2_	296.5	Antibacterial [36]
7	28.461	48,494	Methyl (9E)-9-octadecenoate	C_19_H_36_O_2_	296.5	Anticancer [37], antibacterial, and antioxidant [38,39]
8	28.845	205,937	Stearic acid, methyl ester	C_19_H_38_O_2_	298.5	Antibacterial [40]
9	29.235	32,457	9-Octadecenal, (Z)-	C_18_H_34_O_2_	266.5	Antibacterial and antifungal [41]
10	29.357	17,037	cis,cis-Linoleic acid	C_18_H_32_O_2_	280.45	Antimicrobial, antioxidant [42]

**Table 2 biomedicines-11-02832-t002:** Summary of the properties, formulas, molecular weights, and retention times for significant compounds identified in CME@ZIF-8 MOFs.

No	RT (m)	Area	Compound Name	MF	Mol. Wt. g/mol	Properties
1	21.722	70,816	Methyl 8-(2-octyl cyclopropyl) octanoate	C_20_H_38_O_2_	310.5	Antibacterial and antioxidant [43]
2	23.817	698,232	Hexadecanoic acid, methyl ester	C_17_H_34_O_2_	270.5	Antibacterial [31]
3	27.648	5650	1,2-Cyclohexanedimethanol	C_8_H_16_O_2_	144.21	--
4	27.879	5229	1,E-11,Z-13-Octadecatriene	C_16_H_32_O_2_	256.42	Anticancer [44]
5	28.210	67,605	Methyl (9E,12E)-9,12-octadecadienoate	C_19_H_34_O_2_	294.5	Antibacterial, antioxidant [35]
6	28.333	365,536	Oleic acid, methyl ester	C_19_H_36_O_2_	296.5	Antibacterial, antimicrobial [36]
7	28.459	128,105	7-Hexadecenoic acid, methyl ester, (Z)-	C_17_H_32_O_2_	268.4	Antibacterial and antioxidant [45]
8	28.842	526,375	Stearic acid, methyl ester	C_19_H_38_O_2_	298.5	Antibacterial [46]
9	29.017	14,779	(1-Methyl-1-propylpentyl) benzene	C_15_H_24_	204.35	--
10	29.266	18,651	Nopyl acetate	C_13_H_20_O_2_	208.30	--

## Data Availability

Data are available upon request.

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
