# Peer review of "Advancing Nanoscale Science: Synthesis and Bioprinting of Zeolitic Imidazole Framework-8 for Enhanced Anti-Infectious Therapeutic Efficacies"

_biomedicines, 2023, doi:10.3390/biomedicines11102832_

Round 1

Reviewer 1 Report

The authors put a lot of work into the research, however article needs some revisions before publication.

1) S. aureus and E. coli – should be in italic

2) 2.2. Plant Extract Preparation

“To prepare the plant extraction, we used 25 g of plant powder in a 1000 mL beaker with 500 mL of distilled water” – please add in brackets plant name. What is the origin of plant powder? Please provide supplier.

3) 2.6. Antimicrobial Tests, 2.6.1. Disc diffusion method:

please provide the names of the bacterial strains and bacteria supplier,

“different concentrations of ZIF-8, CME@ZIF-8, CME and ciprofloxacin were applied to 6 mm diameter filter paper disks” – please list these concentrations,

please specify on which media and agars bacteria were cultured and grown,

4) 2.6.2. Minimum inhibitory Concentration

“In this study, CME@ZIF-8 showed the minimum concentration when compared with ZIF-8, CME and ciprofloxacin against both positive and negative strain.”- instead of positive and negative strain please used the bacteria name or gram-positive and gram-negative bacteria.

5) Page 7, line 217

“XRD analysis revealed that CME@ZIF-8 particles had high crystallinity.” - What does the author mean when he writes: high crystallinity.

6) Please  elaborate the EDX section.

Reviewer 2 Report

The paper entitled „Advancing Nanoscale Science: Synthesis and Bioprinting of ZIF-8 MOF for Enhanced Anti-infectious Therapeutic Efficacies” focuses on encapsulation of bioactive molecule from the Cordia myxa extract using zeolitic imidazolate framework-8 to synthesize nanocomposite (CME@ZIF-8). The obtained ceramic powders were subjected to X-ray diffraction (XRD), UV-visible spectrophotometry, scanning electron microscopy (SEM), energy-dispersive X-ray spectroscopy (EDX) and Fourier transform infrared radiation (FTIR). The topic of the paper is interesting, but the introduction and the aim of the study should be improved.  Also, the results and conclusions need to be elaborated.

I would like to recommend the publication of the paper for publication after addressing some issues as follows:

1. Please, avoid using abbreviations in the title and abstract;

2.               The introduction section must be elaborated. More focused literature review on similar research including what has been achieved so far and what is new in the present study is needed;

3.               The aim of the study is missing. It should be accurately and precisely formulated.

4.               The statistical analysis applied in the study is not mentioned. The number of tests conducted is missing.

5.               The letters used in Fig. 2 are too small to be clearly seen.

6.               The English style can be improved. In some places, punctuation and grammar mistakes are detected.

7.               In the conclusion, the authors say that “CME@ZIF-8 can be a strong drug release carrier as it has excellent stability and biocompatibility. How were both stability and biocompatibility ssessed? Moreover, the statement “Small sized CME@ZIF-8 can easily penetrate the bacterial cells and produce ROS…” also needs to be proved by experimental results.

8.               The reference style does not correspond to that required by the journal.

In some places, punctuation and grammar mistakes are detected.

Round 2

Reviewer 1 Report

In my opinion, the article is suitable for publication in its current form.  However, if there is a possibility please improve the quality of Fig. S1. EDX analysis. 

Author Response

Revised Figure 

Reviewer 2 Report

The authors have carefully addressed the reviewer's recommendations. 

None

Author Response

Thanks alot